# Distributionally Robust Model and Metaheuristic Frame for Liner Ships Fleet Deployment

**Mihaela Bukljaš [1,\*], Kristijan Rogić [2] and Vladimir Jerebić [3]**

1 Department of Water Transport, Faculty of Transportation and Traffic Sciences, University of Zagreb, 10000 Zagreb, Croatia
2 Department of Transport Logistics, Faculty of Transportation and Traffic Sciences, University of Zagreb, 10000 Zagreb, Croatia; krogic@fpz.unizg.hr
3 Faculty of Transportation and Traffic Sciences, University of Zagreb, 10000 Zagreb, Croatia; vladimir.jerebic1@optinet.hr
\* Correspondence: mbukljas@fpz.unizg.hr

**Abstract:** The container shipping industry market is very dynamic and demanding, economically, politically, legally, and financially. Considering the high cost of core assets, ever rising operating costs, and the volatility of demand and supply of cargo space, the result is an industry under enormous pressure to remain profitable and competitive. To maximize profits while maintaining service levels and ensuring the smooth flow of cargo, it is essential to make strategic decisions in a timely and optimal manner. Fleet deployment selection, which includes the profile of vessel hire, as well as their capacity and port rotation, is one of the most important strategic and tactical decisions container shipping operators must make. Bearing in mind that maritime business is inherently stochastic and uncertain, the key aims of this paper are to address the problem of fleet deployment under uncertain operating conditions, and to provide an integrated and optimized tool in the form of a mathematical model, metaheuristic algorithm, and computer program. Furthermore, this paper will show that the properties of the provided solutions exceed those offered in the literature so far. Such a solution will provide the shipping operator with a decision tool to best deploy its fleet in a way that responds more closely to real life situations and to meet the maximum demand for cargo space with minimal expense. The final goal is to minimize the operating costs while managing cargo flows and reducing the risks of unfulfilled customer demands.

**Keywords:** liner shipping; fleet deployment; distributional robust optimization; chance constrained optimization; metaheuristics

## 1. Introduction

Global World trade reached 28.5 trillion USD in 2021. This is an increase of 25% compared to 2020 [1]. Maritime transport, being a derivate of global industry and commerce, has contributed 93% of its volume, or 11.08 billion tons, to the transport of goods, and derives 73% of its value from this business. At the same time, container shipping fleets, amounting to a total of 5418 fully cellular ships with a combined capacity of 24,140,554 TEUs, comprises 17.1% of the global maritime transport volume [2].

Due to high financial stakes, the core value of the assets, significant operational expenses (OPEX), and, above all, the fierce competition of the market, the control of container shipping is achieved through three levels: strategic, tactical, and operational [3]. Fleet deployment (vessel fleet deployment—VFD), is a tactical-level concern, the main task of which is to assign ships or fleets to a sequence of ports so as to maximize the revenue.

To optimize the transport of containers and maintain an uninterrupted flow of cargo, container shipping operators have established worldwide service lines. These services enable reliable sailing schedules, continuous delivery, and safe and cost-effective transportation, making goods available at a specific time. However, container line services are

regulated in advance, usually for a planning horizon of three to six months, which means that the network is designed before the demand for delivery and distribution of goods is fully known. For this reason, determining the demand for the cargo space within the line service is a key element in the design of these networks, the primary goal of which is to minimize the total costs while meeting the demand and maximizing profits.

Today, the major cargo shipping lines stretch in the east–west direction, connecting the industrial centers of Europe, North America, and Asia, while the north–south routes reaching South America and Africa are also growing strongly. These lines facilitate the heaviest trade and supply chains for goods globally, and their maintenance, efficiency, and profitability are unimaginable without modern container ships, which require quality infrastructure in the areas they serve.

Determining infrastructure requirements and fleet capacity under conditions of unstable demand is one of the essential tasks in planning and managing these supply chains. Seaborne transportation is a crucial element in transportation planning, especially container transportation, because of its share in total sea transportation. Therefore, the problem of fleet capacity planning and deployment has direct influence on the profitability and competitiveness of a single operator. This problem is even more conspicuous under the conditions of variable demand and insufficient fleet capacities that have become common during the pandemic period.

This paper studies the problem of container ship fleet deployment under uncertain shipment requirements. The aim is to minimize the sum of the cost of chartering the vessel and the operating costs of the route, while controlling the risk of excessive demand for the consignment, i.e., the risk of demand exceeding the ship capacity.

Maritime industry has always presented a challenge for any form of "traditional" optimization due to its innate stochastic nature, the uncertainty of events, and quick changes in business demands and conditions. There is even a commonly accepted term, "maritime endeavor", to describe the particularity of the shipping business. The results and robust model developed and presented in this paper may be applicable to other segments of the shipping industry. However, due to its highly dynamic operations and the stochastic nature of the variables involved, container shipping is most sensitive to decision errors due to unknown data. Therefore, an advanced, distributionally robust model for fleet deployment is essential for the profitable management of container shipping liner services.

## 2. Literature Review

A view of containerized trade and its historical trends since 1996 is presented in Figure 1, revealing continuous growth in trading volumes, with the exception of the years following the 2009 global economic crisis and the recent COVID-19 pandemic shutdowns.

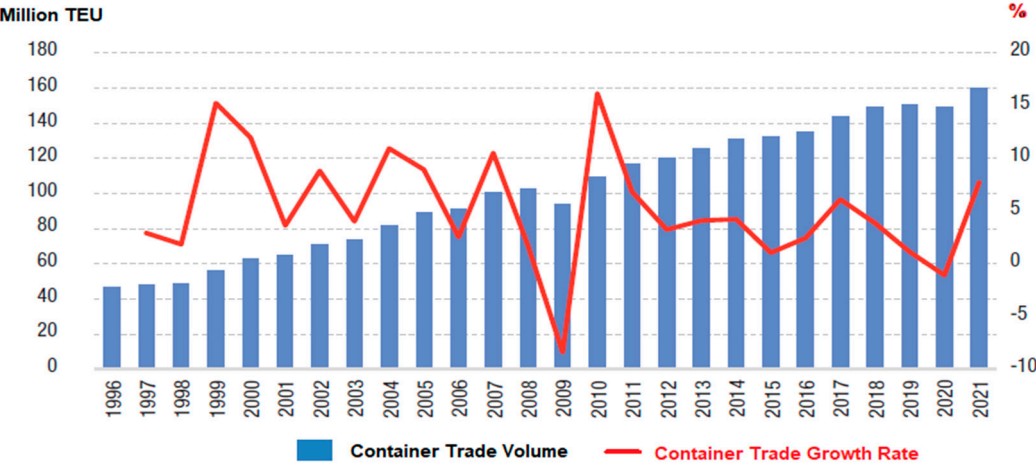

**Figure 1.** Global Containerization Trade (1996–2021), Source [2].

The continuous volume increases, despite the crisis, clearly indicate the importance of containerized modal transport within the supply chains. Its resilience and the previously mentioned properties are today, without question, the major engine for the globalization processes.

However, due to high financial stakes, the core value of the assets, significant operational expenses (OPEX) and above all, fierce competition on the market, the control of the container shipping is done through three levels: strategic, tactical, and operational [3]. The Fleet deployment, (vessel fleet deployment—VFD), being of tactical level has the main task of assigning ship or fleet of ships to the sequence of ports in such a way as to maximize the revenue.

The problem of optimal VFD decision-making is one of the key issues in the shipping industry, as confirmed by the research done by Wang and Meng [4], and Zhang et al. [5]. The VFD problem is usually addressed by deterministic container delivery requirements [3]. Liner shipping could be taken as an example. Receiving and delivering of containers on the shipping routes is the main duty of liner shipping. The shipping company mainly provides container-shipping services with deadlines for worldwide shippers. The key problem of ship fleet deployment is the optimization of the allocation of quantities and types of different ships for each ship service route, to achieve the lowest possible cost by efficient ship management [6].

The VDF problem could be expressed by determining the quantities of different types of vessels to be employed (on or off hired), the quantity of different types of vessels to be deployed on each service route and the number of voyages to be performed on the main haul route within the stochastic environment. To maintain the competitiveness, a predetermined level of shipping service is supposed to be guaranteed. The probability of meeting all shipping requirements serves as an indicator of the level of shipping services. Uncertainty lies mainly in systems from trade policy and oil prices. Noticing the importance of stochastic requirements in the shipping industry; some scientists study the problem of deploying a stochastic fleet of ships [7].

The probability distribution of a random factor must in practice be estimated from historical data and replacing the unknown probability distribution with an estimate may not meet the chance constraint underlying the actual probability distribution. Therefore, this work will utilize a robust distribution framework for the problem of ship fleet deployment under uncertainty sets [8].

Being a relatively young field of operational research, during the last two decades, robust optimization has become largely popular, as a modeling substratum for the protection against variable uncertainties in mathematical optimization [9–20].

As previously said, the VFD problem could be summarized as a task to determine how many vessels to deploy on a respective service route, their capacity structure, and the number of voyages to be completed to minimize the expenses, maximize the profit and maintain the service level along the planned horizon.

The first steps to develop a model for the VDF problem reach back to the early 1990s with the works of Perakis and Jaramillo [21], Cho and Perakis [22] and Powel and Perakis [23]. All based on linear programming. Later, Gelareh and Meng [24], Wang et al. [25], Meng and Wang [26] tried to improve the above.

All the previous research was deterministic in its nature and did not provide an answer to uncertainty demands of VFD. To maintain the service level and the operator's competitiveness, all shipping demands must be met with high probability. However, due to trade policies, economic, political, and financial factor as well as fuel price fluctuation, the container market is inherently stochastic. To capture such uncertainty, numerous works emerged at the beginning of 2000s. The papers by Christiansen et al. [27,28], Meng and Wang [29], Meng et al. [30] and Wang et al. [31] have established the chance constraints to meet the stochastic demands, but under the known probability distribution only. Ng [9,32] introduced works that somehow dealt with the unknown distribution. In practice, the probability distribution of uncertain or random factors must be estimated based upon historical data to replace the unknown probability [33]. Therefore, distributional robust (further DR) framework models are convenient for modeling under uncertain or ambiguous demands.

It is necessary to highlight the works of Delage and Ye [7] for developing the DR approaches, as well as Jiang and Guan [34], Zhang et al. [35] and Sui and Zheng [36]. They have proposed a new distributional robust framework to establish an approximation, cost saving method.

Liner carriers should regularly adjust their shipping networks to respond to their competitors. Also, they should adjust the offer to change the seasonal requirements of customers. This is done by selecting and changing the ship routes. The most modern approaches imply a separation into two separate problems. In these cases, an integrated mathematical model and a mathematical model for repositioning the liner fleet are used, which jointly optimizes the choice of vessels for routes and the costs of moving ships according to their tasks (according to demand). In this way, the simultaneous optimization of the setup and relocation can result in a significant reduction in costs during self-solving business problems [37].

In the study conducted by Dong et al. [38], the combined problem of fleet allocation and inventory management (FDIMP) that occurs in Roll-on Roll-off (Ro-Ro) delivery is considered. In addition to several predetermined trade routes with a series of port calls, many loading and unloading ports possess stocks to be kept within their borders. The current planning practice is to visit all ports every time a trade route is serviced. Here, instead, the authors sought to determine the sailing routes of each voyage along the trade route, that is, which ports to visit, where some ports along the trade route can be bypassed on certain voyages. They proposed a new mixed integer model for this new and more flexible version of FDIMP in Ro-Ro shipping. The flexible model gives much better results than the basic model without any flexibility of arc skipping. The results of their research show the potential economic effect of changing the way of planning, i.e., by introducing the possibility of bypassing ports, which can be possibly obtained if stronger cooperation with shipowners or customers is achieved. Significant cost savings are achieved, from inflexibility to even tight inventory constraints, and significant further savings are also achieved.

The subject of optimization within the shipping industry is of great importance. Even though initial works on this problem date to the 1990s, it was only recently that the researchers began to explain the significant amount of uncertainty that is present in the actual demand for ships [32].

Distribution-robust optimization provides the basis for creating machine-learning models that can derive the sum of related data distributions. This is achieved by enabling the model to mitigate its maximum expected loss among all distributions in the uncertainty segment [9].

The distribution-robust optimization has several stunning advantages. It allows modelers to embed error estimation problems into optimization problems. It therefore results in a more realistic representation of uncertainty and alleviates the course of the optimizer characteristic of classical stochastic programming. The distribution-robust optimization problems can be often solved accurately and in polynomial time—in stark contrast to unsolvable approximate models obtained by discretizing stochastic problems adapted to a single nominal distribution. The distribution robust optimization models can adapt to the size of industry-relevant problems and are already used in a few fields of practice, including vehicle routing, fleet management, portfolio selection, revenue management, scheduling, environmental policies, smart grid management, and so on [10].

The distributional robust model is quite attractive for a myriad of reasons. The first reason to use this model is fidelity. The distributional robust model is more reliable than its counterparts since they find the presence of distributional uncertainties. They also have great advantages when it comes to information about the type and magnitude of estimation errors. Using the distributional robust model, it is easy to manage all the expectations since the solutions of this model will display risks that fall below the worst-case optimal risk when the uncertainty set has an unknown true distribution [12]. This model has a very high-performance guarantee that other models do not have, and it also has an outstanding computational traceability.

In this paper, the distributional robust model was used for planning fleet capacities and deployment in container shipping. If compared to the above mentioned basic and advanced papers, mathematical model is more flexible, enabling the circular routes, which is not the case in [36,39]. Also, the models provided do not offer complete heuristic solution to problems that may be used for a broad spectrum of shipping deployment problems. Pseudo code and MIP model have a great benefit of being easily processed by several, readily available computer programs, and their modifications do not require either excessive time-consuming labor nor computational power. Such approach in the modelling process enables control of demand overflow risks, where there is a presumption that the ship capacity demand probability distributions are indeterminate. The works previously presented in literature [21,31,34,40,41] cannot offer such an option. A robust distribution framework was also used to establish a new approximation method that could significantly save costs. The paper contribution mainly covers the following [11]:

- formulation of distributional robust model;
- demonstration that this approximation is more general than the most modern method.
- development of metaheuristic frame.
- development of computer-based program for solving the fleet deployment problem under uncertainty within acceptable CPU time.

## 3. Model Construction

The aim in this paper is to develop an efficient optimization model which regards the repositioning of container ships between the existing line services based on the robust optimization techniques and methods. This model contains a defined and developed mathematical model of robust optimization for vessel/fleet repositioning between the existing line services under certainty conditions. In this model the efficient evolutionary algorithm has been developed and implemented to find the optimal acceptable solution to the problem of fleet/vessel repositioning. By implementing this model and algorithm it is possible to obtain higher profit maximization than with traditional discrete optimization methods.

As said, to achieve the best feasible robust solution to fleet deployment problem, as close to optimal as possible, the authors have adopted the mixed approach. The mathematical model, metaheuristic algorithm supported by computer simulation in CPLEX MATLAB.

Based on the defined goal of this paper, scientific contributions will be obtained, which are manifested through the following hypotheses:

- it is possible to develop an innovative mathematical model of robust fleet deployment that will offer feasible solution(s) fast and cost-efficiently;
- it is possible to create efficient metaheuristic algorithm(s) in the decision-support system in solving the problem of fleet deployment.

### 3.1. Problem Description

To develop a distributional robust (further DR), chance-constrained (further CC) model for fleet deployment [4] the first step is the problem description.

Let $K = \{1, \ldots, K\}$ denote the set of vessel types, available to container shipping liner company (further - operator) and $k$ the specific ship type. $V_k$ is the ship capacity in TEUs (*Twenty-foot Equivalent Unit* is the capacity unit in intermodal transport and represents ISO steel container of dimensions $6.1 \times 2.44 \times 2.59$ m and $38.5$ m$^3$ of volume) for Ship $k$. An operator should determine the finite number of ships $k \in K$ and deploy them on routes $r \in R$, while following the sailing schedule. Beside their own ships, an operator's broker may charter the available ships from the market for the duration of the planning horizon. The charter rate for Ship $k$ is $C_k^{in}$ (USD/ship). $N_k^{max}$ and $NCI_k^{max}$ denote the number of ships $k$, either owned or chartered. Such a number is often limited due to expenses involved or the ships availability on the open charter market.

*3.2. Scenario for Container Shipment Demand*

The uncertainty for container shipping demand (further demand only) is embedded into the model by finite number of demand scenarios. In any of scenarios, the demand values for each pair origin-destination ports (O-D pair) are set for the duration of the planning horizon. The assumption is that the demand of certain O-D pairs throughout the planning horizon is a discrete variable $\zeta^{od}$ (($o, d$) $\in W$). $\zeta^{od}$ is limited to a certain range under the known probability values. Let $s \in S = \{1, 2, 3, \ldots, S\}$ be a set of demand scenarios. The realization of random variable $\zeta^{od}$ within scenario $s \in S$ is denoted by $\omega_s^{od}$. The probability of $s \in S$ is $P_s = \sum_{s=1}^{S} P_s = 1$.

*3.3. Model Buildup*

As mentioned before, maritime business is uncertain and stochastic and therefore, it is necessary to provide a model for fleet deployment which will be close to real living conditions. The main goal here is to cut down the costs to a minimum level while managing cargo demand risk. In this case a stochastic dynamic program will not be used, but in order to solve this multi region system, the distributional robust optimization has been deployed which can incorporate demand temporal dependence motivated by real data. In the first part, the charter income will be set while taking into consideration the demand realization. The model will be used to structure the expenses such as cargo handling costs, vessel operating costs, and investments. Based on all the given costs, an optimization model which has the objective of maximizing the income will be obtained.

3.3.1. Decision Variables

$n_{kr}^{own}$ = number of owned ships $k$ ($k \in K$) assigned to route $r$ ($r \in R$);
$n_{kr}^{in}$ = number of chartered ships $k$ ($k \in K$) assigned to route $r$ ($r \in R$);
$x_{kr}$ = number of voyages by ship $k$ ($k \in K$) on route $r$ ($r \in R$); and
$z_s^{h^{od}}$ = number of container units carried by ships $h^{od} \in H^{od}$ between O-D pair ($o, d$) $\in W$ as per demand scenario.

3.3.2. Parameters

$c_{kr}$ = operating expenses of ship $k$ on route $r$ per voyage (USD/Voyage);
$c^{h^{od}}$ = cargo handling expenses O-D port pair ($o, d$) $\in W$ on container flow $h^{od} \in H^{od}$ (USD/TEU);
$C_k^{in}$ = charter rate for hired in ship $k$ during planning horizon (USD/Ship);
$C_k^{out}$ = charter rate for hired out ship $k$ during planning horizon (USD/Ship);
$f^{od}$ = freight rate between O-D port pair ($o, d$) $\in W$ (USD/TEU);
$N_k^{max}$ = number of owned ships of type $k$;
$NCI_k^{max}$ max = number of available ships of type $k$ on charter market;
$c_{kr}$ = number of sailing days for ship $k$ on route $r$ (in days);
$T$ = length of planning horizon (short-term—6 months);
$U_s^{od}$ = minimum number of TEUs for shipment between O-D port pair ($o, d$) $\in W$ by scenario $s$ (TEU);
$V_k$ = container capacity of ship $k$ (TEU);
$N_r$ = minimum number of voyages on route $r$ during the planning horizon to keep the requested level of service (service schedule); and
$\rho_{ir}^{h^{od}}$ = binary coefficient, shows if container flow path $h^{od} \in H^{od}$
    contains the subset of route $r$ ($\rho_{ir}^{h^{od}} = 1$) or not ($\rho_{ir}^{h^{od}} = 0$).

### 3.4. Optimization Model

The operator's revenue generally has two sources. The income of chartering out the vessel to other operators and the income from the container freight. Charter income is set by

$$\sum_{k \in K} c_k^{out} \left( N_k^{max} - \sum_{r \in R} n_{kr}^{own} \right) \tag{1}$$

While taking into consideration scenario *s* and demand realization $\omega_s^{od}$ for O-D port pair $(o, d) \in W$, the container freight income is given by equation:

$$\sum_{(o,d) \in W} \sum_{h^{od} \in H^{od}} f^{od} z_s^{h^{od}} \left( \omega_s^{od} \right) \tag{2}$$

The expenses may be structured as simple as: cargo handling costs, vessel operating costs (OPEX) and investments into chartered vessels. Cargo handling costs include container loading, discharging or restoring expenses.

$$\sum_{(o,d) \in W} \sum_{h^{od} \in H^{od}} c^{h^{od}} z_s^{h^{od}} \left( \omega_s^{od} \right) \tag{3}$$

Let $c_{kr}$ denote the voyage operating cost for ship *k* on route *r*. The $c_{kr}$ contains fuel cost, daily labour cost, harbour and light dues. Ships OPEX plus charter fees may be expressed as follows:

$$\sum_{r \in R} \sum_{k \in K} c_{kr} x_{kr} + \sum_{k \in K} c_k^{in} n_{kr}^{in} \tag{4}$$

Therefore, the optimization model with the objective of maximizing the income may be obtained as per Equation (5):

$$Z_0 = max \sum_{s \in S} P_s \times \left( \begin{array}{c} \sum_{k \in K} c_k^{out} \left( N_k^{max} - \sum_{r \in R} n_{kr}^{own} \right) \\ + \sum_{(o,d) \in W} \sum_{h^{od} \in H^{od}} f^{od} z_s^{h^{od}} \left( \omega_s^{od} \right) - \sum_{(o,d) \in W} \sum_{h^{od} \in H^{od}} c^{h^{od}} z_s^{h^{od}} \left( \omega_s^{od} \right) \\ - \sum_{r \in R} \sum_{k \in K} c_{kr} x_{kr} + \sum_{k \in K} c_k^{in} n_{kr}^{in} \end{array} \right) \tag{5}$$

Subject to:

$$\sum_{r \in R} n_{kr}^{own} \leq N_k^{max} \tag{6}$$

$$\sum_{r \in R} n_{kr}^{in} \leq NCI_k^{max} \tag{7}$$

$$x_{kr} \leq \left( n_{kr}^{own} + n_{kr}^{in} \right) * \left[ \frac{T}{t_{kr}} \right], \ \forall r \in R, \ \ \forall k \in K \tag{8}$$

$$\sum_{k \in K} x_{kr} \geq N \ \ \ \forall r \in R \tag{9}$$

$$\sum_{k \in K} x_{kr} V_k \geq \sum_{(o,d) \in W} \sum_{h^{od} \in H^{od}} \rho_{ir}^{h^{od}} z_s^{h^{od}} \left( \omega_s^{od} \right) i = 1, \ldots, m_r \forall r \in R, \ \ \forall s \in S \tag{10}$$

$$u_s^{od} \leq \sum_{h^{od} \in H^{od}} z_s^{h^{od}} \left( \omega_s^{od} \right) \leq \xi^{od} \left( \omega_s^{od} \right), \ \forall (o,d) \in W, \ \ \forall s \in S \tag{11}$$

$$n_{kr}^{own}, \ n_{kr}^{in}, \ x_{kr} \in Z^+ U\{0\}, \ \forall r \in R, \ \ \forall k \in K \tag{12}$$

$$z_s^{h^{od}} \geq 0, \ \forall h^{od} \in H^{od}, \ \ \forall s \in S \tag{13}$$

Equation (5) is the objective function for the optimization model. It is also known as the model of the expected value. Constraint sets (6) and (7) ensure that the number of owned and chartered ships does not exceed the available number of vessels. Expression (8) returns with the max number of voyages for ships of *k* type on route *r*, where $\left[ \frac{T}{t_{kr}} \right]$ denotes

the maximal integer smaller than or equal to $\frac{T}{t_{kr}}$. Constraint (9) specifies the number of voyages on route $r$ needed to maintain the sailing schedule.

The left side in constraint (10) represents the total transport capacity of a ship deployed on liner route $r \in R$; while the right side is the total number of containers being transported on any section $i$ of route $r \in R$, including the container loaded in the previous ports (still on board) and those loaded or transshipped in port $p_r^i$. Therefore, this constraint ensures that the number of containers offered for transport does not exceed the capacity of deployed ships.

Traditionally, liner operators join conferences or form alliances to gain market access, increase operational flexibility, and better cope with the competition or even economic crisis. As a result, due to the signed obligation, a contractual number of containers must be shipped out from the ports on route, while the rest is optional. The right side of constraint (11) is the demand realization for O-D port pair $(o, d) \in W$ as per scenario $s$. The right side of inequality ensures that the number of carried containers cannot exceed the demand, while at the same time, the left side makes sure that the agreed number of units must be shipped out.

Constraints (12) and (13) define the range of decision variables. Finally, the objective function could be written as follows:

$$
\begin{aligned}
Z_1 = min \sum_{r \in R} \sum_{k \in K} \left( c_k^{out} n_{kr}^{own} + c_{kr} x_{kr} + c_k^{in} n_{kr}^{in} \right) \\
+ \sum_{s \in S} \sum_{(o,d) \in W} \sum_{h^{od} \in H^{od}} P_s \left( c^{h^{od}} - f^{od} \right) z_s^{h^{od}} \left( \omega_s^{od} \right)
\end{aligned}
\tag{14}
$$

s.t. constraints (6)–(12).

### 3.5. Robust Optimization Model

Robust optimization (further RO) has brought several solutions gradually less sensitive to the realization of data from the set of given scenarios. RO includes two major sets of variables: design and control variables. The design variables are the decision variables whose optima are not subject to realization of uncertain parameters. As such, they cannot be adjusted upon the realization of certain data that have been observed, while control variables are subject of adjustment after the uncertain parameters are identified. Their optima depend both on the realization of uncertain parameters and on the optimal value of the design variables.

Let $x \in R^{n_1}$ denote the vector of decision variables, and $y \in R^{n_2}$ the vector of control variables. The general model of linear programing (further LP) has the following structure:

$$
min c^T x + d^T y
\tag{15}
$$

s.t.

$$
Ax = b
\tag{16}
$$

$$
Bx + Cy = e
\tag{17}
$$

$$
x, y \geq 0
\tag{18}
$$

Constraint (16) marks the fixed, structural coefficients, while (17) denotes control constraints, where coefficients are subject to uncertainty. The problems, modeled by RO include scenario $s \in S = \{1, 2, 3, \ldots, S\}$. Furthermore, the set of controlled variables for each scenario $s \in S$ and set $\{\varepsilon\_1, \varepsilon\_2, \ldots, \varepsilon\_s\}$ of error vectors. Error vectors measure the allowed infeasibility or reduced feasibility level in control constraints of scenario $s \in S$.

Then, the general form of robust optimization model (ROM) has the following structure:

$$
min \sigma(x, y_1, y_2, \ldots, y_s) + \overline{\omega} \rho(\varepsilon_1, \varepsilon_2, \ldots, \varepsilon_s)
\tag{19}
$$

s.t.

$$
Ax = b
\tag{20}
$$

$$B_s x + C_s y_s + \varepsilon_s = e \tag{21}$$

$$x, y_s \geq 0 \tag{22}$$

Since ROM considers several scenarios, the objective function from Equation (15) becomes random variable of the value $\zeta_s = c^T x + d^T y_s$, with probability $p_s$.

The next step is to introduce an objective weight coefficient $\omega$, with the purpose of obtaining the range of solutions that may vary during the search for the optimum ROM solution. They are the tradeoff solution to gain robustness of the model. Expressions $\sigma(x, y_1, y_2, \dots, y_s)$ and $(\varepsilon_1, \varepsilon_2, \dots, \varepsilon_s)$, proposed by Yu and Li [39] are given by the following equations:

$$\sigma(x, y_1, y_2, \dots, y_s) = \sum_{s \in S} p_s \zeta_s + \lambda \sum_{s \in S} p_s \left| \zeta_s - \sum_{\acute{s} \in S} p_{\acute{s}} \zeta_{\acute{s}} \right| \tag{23}$$

$$\rho(\varepsilon_1, \varepsilon_2, \dots, \varepsilon_s) = \sum_{s \in S} p_s max\{0, \varepsilon_s\} \tag{24}$$

The same authors have developed an efficient model for solving the absolute error from (23) within the frame designed as follows:

$$min \sum_{s \in S} p_s \zeta_s + \lambda \sum_{s \in S} p_s \left[ \left( \zeta_s - \sum_{\acute{s} \in S} p_{\acute{s}} \zeta_{\acute{s}} \right) + 2\vartheta_s \right] + \overline{\omega} \sum_{s \in S} p_s \varepsilon_s \tag{25}$$

$$Ax = b \tag{26}$$

$$B_s x + C_s y_s + \varepsilon_s = e, \ \forall s \in S \tag{27}$$

$$\zeta_s - \sum_{\acute{s} \in S} p_{\acute{s}} \zeta_{\acute{s}} + \vartheta_s \geq 0, \ \forall s \in S \tag{28}$$

$$x, y_s, \ \varepsilon_s \ \vartheta_s \geq 0, \ \forall s \in S, \ \forall s \in S \tag{29}$$

The above introduction was essentially made to simplify the distinction between control and design variables proposed in this paper. Variables $n_{kr}^{own}$, $n_{kr}^{in}$ and $x_{kr}$ are design variables and $z_s^{h^{od}}$ is control variable.

Following $\zeta_s$ from (25) and objective function (14), the proposed fleet deployment model is:

$$\begin{aligned} &\sum_{r \in R} \sum_{k \in K} (c_k^{out} n_{kr}^{own} + c_{kr} x_{kr} + c_k^{in} n_{kr}^{in}) + \\ &\sum_{s \in S} \sum_{(o,d) \in W} \sum_{h^{od} \in H^{od}} P_s \left( c^{h^{od}} - f^{od} \right) z_s^{h^{od}} \left( \omega_s^{od} \right) \end{aligned} \tag{30}$$

Therefore, new ROM is formulated as:

$$\begin{aligned} min \ &\sum_{r \in R} \sum_{k \in K} (c_k^{out} n_{kr}^{own} + c_{kr} x_{kr} + c_k^{in} n_{kr}^{in}) + \\ &\sum_{s \in S} \sum_{(o,d) \in W} \sum_{h^{od} \in H^{od}} P_s \left( c^{h^{od}} - f^{od} \right) z_s^{h^{od}} \left( \omega_s^{od} \right) + \\ &\lambda \sum_{s \in S} P_s \left[ \begin{array}{c} \sum_{(o,d) \in W} \sum_{h^{od} \in H^{od}} \left( c^{h^{od}} - f^{od} \right) z_s^{h^{od}} \left( \omega_s^{od} \right) - \\ \sum_{\acute{s} \in S} \sum_{(o,d) \in W} \sum_{h^{od} \in H^{od}} P_{\acute{s}} \left( c^{h^{od}} - f^{od} \right) z_{\acute{s}}^{h^{od}} \left( \omega_{\acute{s}}^{od} \right) + 2\vartheta_s \end{array} \right] + \overline{\omega} \sum_{s \in S} \sum_{r \in R} \sum_{i=1}^{m_r} p_s \varepsilon_s^{ir} \end{aligned} \tag{31}$$

s.t. (6)–(8), (11), (12) and:

$$\sum_{k \in K} x_{kr} V_k + \varepsilon_s^{ir} \geq \sum_{(o,d) \in W} \sum_{h^{od} \in H^{od}} \rho_{ir}^{h^{od}} z_s^{h^{od}} \left( \omega_s^{od} \right) i = 1, \ \dots, \ m_r \forall r \in R, \quad \forall s \in S \tag{32}$$

$$
\begin{bmatrix}
\displaystyle\sum_{(o,d)\in W}\ \sum_{h^{od}\in H^{od}}\left(c^{h^{od}}-f^{od}\right)z_s^{h^{od}}\left(\omega_s^{od}\right)- \\[2mm]
\displaystyle\sum_{\acute{s}\in S}\ \sum_{(o,d)\in W}\ \sum_{h^{od}\in H^{od}}P_{\acute{s}}\left(c^{h^{od}}-f^{od}\right)z_{\acute{s}}^{h^{od}}\left(\omega_{\acute{s}}^{od}\right)+\vartheta_s
\end{bmatrix}\geq 0 \tag{33}
$$

$$
\varepsilon_s^{ir}\geq 0,\ \vartheta_s\geq 0\geq 0 \tag{34}
$$

The constraint set (32) are control constraints for determination of container flow assignment for each section $i$ of route $r$, as per certain scenarios $s$. If total capacity of ships deployed on route $r$ (left side) is bigger than the assigned container flow for observed section $i$ on route $r$, then the aberration is $\varepsilon_s^{ir}=0$. Otherwise, $\varepsilon_s^{ir}=\sum_{(o,d)\in W}\sum_{h^{od}\in H^{od}}\rho_{ir}^{h^{od}}z_s^{h^{od}}(\omega_s^{od})-\sum_{k\in K}x_{kr}V_k$ and this shows that the ships are not loaded up to capacity. That way we have an infeasible solution.

The robust optimization model is by its nature an integer model of LP and could be solved by numbers of the existing solvers, such as CPLEX, MATLAB, Gurobi and such. Therefore, the following two novel hypotheses are proposed that differ from the previously developed models, all with the aim of maximizing the cargo intake, hence increasing the overall revenue:

**Proposition 1.** *The variance* $\sum_{s\in S}p_s[(\zeta_s-\sum_{\acute{s}\in S}p_{\acute{s}}\zeta_{\acute{s}})+2\vartheta_s]$ *in optimization model (31) decreases if value of $\lambda$ increases.*

**Proof.** Let us assume that $\lambda^1<\lambda^2$ and $x^1,\ y_s^1,\ \varepsilon_s^1,\ \vartheta_s^1(\forall s\in S)$ and $x^2,\ y_s^2,\ \varepsilon_s^2,\ \vartheta_s^2(\forall s\in S)$ are the optimal solution for $x,\ y_s,\varepsilon_s,\vartheta_s\ (\forall s\in S)$ from the optimization model shown by Equation (25). Moreover $\lambda^1,\ \lambda^2$ are being respectively assigned as well. Objective functions of optimization models $\lambda^1$ and $\lambda^2$ are expressed with $Z\big(x^1,\ y_s^1,\ \varepsilon_s^1,\ \vartheta_s^1\big)\big|_{\lambda=\lambda^1}$ and $Z\big(x^2,\ y_s^2,\ \varepsilon_s^2,\ \vartheta_s^2\big)\big|_{\lambda=\lambda^2}$, respectively as follows.

$$
Z\big(x^1,\ y_s^1,\ \varepsilon_s^1,\ \vartheta_s^1\big)\big|_{\lambda=\lambda^1}=\sum_{s\in S}p_s\zeta_s^1+\lambda^1\sum_{s\in S}p_s\left[\left(\zeta_s^1-\sum_{\acute{s}\in S}p_{\acute{s}}\zeta_{\acute{s}}^1\right)+2\vartheta_s^1\right]+\overline{\omega}\sum_{s\in S}p_s\varepsilon_s^1 \tag{35}
$$

$$
Z\big(x^2,\ y_s^2,\ \varepsilon_s^2,\ \vartheta_s^2\big)\big|_{\lambda=\lambda^2}=\sum_{s\in S}p_s\zeta_s^2+\lambda^2\sum_{s\in S}p_s[(\zeta_s^2-\sum_{\acute{s}\in S}p_{\acute{s}}\zeta_{\acute{s}}^1)+2\vartheta_s^2]+\overline{\omega}\sum_{s\in S}p_s\varepsilon_s^2 \tag{36}
$$

In addition, $Z\big(x^1,\ y_s^1,\ \varepsilon_s^1,\ \vartheta_s^1\big)\big|_{\lambda=\lambda^2}$ and $Z\big(x^2,\ y_s^2,\ \varepsilon_s^2,\ \vartheta_s^2\big)\big|_{\lambda=\lambda^1}$ are given as:

$$
Z\big(x^1,\ y_s^1,\ \varepsilon_s^1,\ \vartheta_s^1\big)\big|_{\lambda=\lambda^2}=\sum_{s\in S}p_s\zeta_s^1+\lambda^2\sum_{s\in S}p_s\left[\left(\zeta_s^1-\sum_{\acute{s}\in S}p_{\acute{s}}\zeta_{\acute{s}}^1\right)+2\vartheta_s^1\right]+\overline{\omega}\sum_{s\in S}p_s\varepsilon_s^1 \tag{37}
$$

$$
Z\big(x^2,\ y_s^2,\ \varepsilon_s^2,\ \vartheta_s^2\big)\big|_{\lambda=\lambda^1}=\sum_{s\in S}p_s\zeta_s^2+\lambda^1\sum_{s\in S}p_s\left[\left(\zeta_s^2-\sum_{\acute{s}\in S}p_{\acute{s}}\zeta_{\acute{s}}^1\right)+2\vartheta_s^2\right]+\overline{\omega}\sum_{s\in S}p_s\varepsilon_s^2 \tag{38}
$$

Therefore:

$$
Z\big(x^1,\ y_s^1,\ \varepsilon_s^1,\ \vartheta_s^1\big)\big|_{\lambda=\lambda^1}\leq Z\big(x^2,\ y_s^2,\ \varepsilon_s^2,\ \vartheta_s^2\big)\big|_{\lambda=\lambda^1} \tag{39}
$$

$$
Z\big(x^2,\ y_s^2,\ \varepsilon_s^2,\ \vartheta_s^2\big)\big|_{\lambda=\lambda^2}\leq Z\big(x^1,\ y_s^1,\ \varepsilon_s^1,\ \vartheta_s^1\big)\big|_{\lambda=\lambda^2} \tag{40}
$$

Summarizing both (39) and (40), we could write (41):

$$
Z\big(x^1,\ y_s^1,\ \varepsilon_s^1,\ \vartheta_s^1\big)\big|_{\lambda=\lambda^1}+Z\big(x^2,\ y_s^2,\ \varepsilon_s^2,\ \vartheta_s^2\big)\big|_{\lambda=\lambda^2}\leq Z\big(x^2,\ y_s^2,\ \varepsilon_s^2,\ \vartheta_s^2\big)\big|_{\lambda=\lambda^1}+Z\big(x^1,\ y_s^1,\ \varepsilon_s^1,\ \vartheta_s^1\big)\big|_{\lambda=\lambda^2} \tag{41}
$$

If (A-1)–(A-4) are inserted into (41) the new inequation is:

$$
(\lambda^1-\lambda^2)\left\{\sum_{s\in S}p_s\left[\left(\zeta_s^1-\sum_{\acute{s}\in S}p_{\acute{s}}\zeta_{\acute{s}}^1\right)+2\vartheta_s^1\right]-\sum_{s\in S}p_s\left[\left(\zeta_s^2-\sum_{\acute{s}\in S}p_{\acute{s}}\zeta_{\acute{s}}^2\right)+2\vartheta_s^2\right]\right\}\leq 0 \tag{42}
$$

From the initial assumption $\lambda^1 < \lambda^2$, the following is easily derived:

$$\sum_{s\in S} p_s\left[\left(\zeta_s^1 - \sum_{\acute{s}\in S} p_{\acute{s}}\zeta_{\acute{s}}^1\right) + 2\vartheta_s^1\right] \geq \sum_{s\in S} p_s\left[\left(\zeta_s^2 - \sum_{\acute{s}\in S} p_{\acute{s}}\zeta_{\acute{s}}^2\right) + 2\vartheta_s^2\right] \tag{43}$$

Therefore, $\left.Var\right|_{\lambda=\lambda^1} = \left.Var\right|_{\lambda=\lambda^2}$. $\square$

Similarly to the above, the next hypothesis may be proposed.

**Proposition 2.** *The mean value of function $\sum_{s\in S} p_s\varepsilon_s$ in optimization model (31) will decrease if the objective weight coefficient $\overline{\omega}$ increases.*

**Proof.** The following proof methodology is like the methodology used to prove Proposition 1. $\square$

Let us assume that $\overline{\omega}^1 < \overline{\omega}^2$, $x^1$, $y_s^1$, $\varepsilon_s^1$, $\vartheta_s^1(\forall s \in S)$ and $x^2$, $y_s^2$, $\varepsilon_s^2$, $\vartheta_s^2(\forall s \in S)$ are optimal solution for $x, y_s, \varepsilon_s, \vartheta_s$ $(\forall s \in S)$ from optimization model shown by Equation (25) and respectively connected to $\overline{\omega}^1$ and $\overline{\omega}^2$. The objective functions of optimization models, denoted $\left.Z\left(x^1, y_s^1, \varepsilon_s^1, \vartheta_s^1\right)\right|_{\overline{\omega}=\overline{\omega}^1}$ and $\left.Z\left(x^2, y_s^2, \varepsilon_s^2, \vartheta_s^2\right)\right|_{\overline{\omega}=\overline{\omega}^2}$, may respectively be shown by the equations below:

$$\left.Z(x^1, y_s^1, \varepsilon_s^1, \vartheta_s^1)\right|_{\overline{\omega}=\overline{\omega}^1} = \sum_{s\in S} p_s\zeta_s^1 + \lambda\sum_{s\in S} p_s\left[\left(\zeta_s^1 - \sum_{\acute{s}\in S} p_{\acute{s}}\zeta_{\acute{s}}^1\right) + 2\vartheta_s^1\right] + \overline{\omega}^1\sum_{s\in S} p_s\varepsilon_s^1 \tag{44}$$

$$\left.Z(x^2, y_s^2, \varepsilon_s^2, \vartheta_s^2)\right|_{\overline{\omega}=\overline{\omega}^2} = \sum_{s\in S} p_s\zeta_s^2 + \lambda\sum_{s\in S} p_s\left[\left(\zeta_s^2 - \sum_{\acute{s}\in S} p_{\acute{s}}\zeta_{\acute{s}}^2\right) + 2\vartheta_s^2\right] + \overline{\omega}^2\sum_{s\in S} p_s\varepsilon_s^2 \tag{45}$$

In addition, $\left.Z\left(x^1, y_s^1, \varepsilon_s^1, \vartheta_s^1\right)\right|_{\overline{\omega}=\overline{\omega}^2}$ and $\left.Z\left(x^2, y_s^2, \varepsilon_s^2, \vartheta_s^2\right)\right|_{\overline{\omega}=\overline{\omega}^1}$ are given as:

$$\left.Z(x^1, y_s^1, \varepsilon_s^1, \vartheta_s^1)\right|_{\overline{\omega}=\overline{\omega}^2} = \sum_{s\in S} p_s\zeta_s^1 + \lambda\sum_{s\in S} p_s\left[\left(\zeta_s^1 - \sum_{\acute{s}\in S} p_{\acute{s}}\zeta_{\acute{s}}^1\right) + 2\vartheta_s^1\right] + \overline{\omega}^2\sum_{s\in S} p_s\varepsilon_s^1 \tag{46}$$

$$\left.Z(x^2, y_s^2, \varepsilon_s^2, \vartheta_s^2)\right|_{\overline{\omega}=\overline{\omega}^1} = \sum_{s\in S} p_s\zeta_s^2 + \lambda\sum_{s\in S} p_s\left[\left(\zeta_s^2 - \sum_{\acute{s}\in S} p_{\acute{s}}\zeta_{\acute{s}}^2\right) + 2\vartheta_s^2\right] + \overline{\omega}^1\sum_{s\in S} p_s\varepsilon_s^2 \tag{47}$$

Therefore:

$$\left.Z(x^1, y_s^1, \varepsilon_s^1, \vartheta_s^1)\right|_{\overline{\omega}=\overline{\omega}^1} \leq \left.Z(x^2, y_s^2, \varepsilon_s^2, \vartheta_s^2)\right|_{\overline{\omega}=\overline{\omega}^1} \tag{48}$$

$$\left.Z(x^2, y_s^2, \varepsilon_s^2, \vartheta_s^2)\right|_{\overline{\omega}=\overline{\omega}^2} \leq \left.Z(x^1, y_s^1, \varepsilon_s^1, \vartheta_s^1)\right|_{\overline{\omega}=\overline{\omega}^2} \tag{49}$$

Adding both sides of (48) and (49), results in (50):

$$\left.Z(x^1, y_s^1, \varepsilon_s^1, \vartheta_s^1)\right|_{\overline{\omega}=\overline{\omega}^1} + \left.Z(x^2, y_s^2, \varepsilon_s^2, \vartheta_s^2)\right|_{\overline{\omega}=\overline{\omega}^2} \leq \left.Z(x^2, y_s^2, \varepsilon_s^2, \vartheta_s^2)\right|_{\overline{\omega}=\overline{\omega}^1} + \left.Z(x^1, y_s^1, \varepsilon_s^1, \vartheta_s^1)\right|_{\overline{\omega}=\overline{\omega}^2} \tag{50}$$

Inserting Equations (44)–(47) into Equation (50) results in:

$$\overline{\omega}^1\sum_{s\in S} p_s\varepsilon_s^1\overline{\omega}^2\sum_{s\in S} p_s\varepsilon_s^2 \leq \overline{\omega}^2\sum_{s\in S} p_s\varepsilon_s^1 + \overline{\omega}^1\sum_{s\in S} p_s\varepsilon_s^2 \tag{51}$$

And that could be shortened as:

$$(\overline{\omega}^1 - \overline{\omega}^2)\left\{\sum_{s\in S} p_s\varepsilon_s^1 - \sum_{s\in S} p_s\varepsilon_s^2\right\} \leq 0 \tag{52}$$

$$\sum_{s\in S} p_s\varepsilon_s^1 \geq \sum_{s\in S} p_s\varepsilon_s^2 \tag{53}$$

## 4. Results

For the computational model of the proposed fleet deployment robust optimization model, the authors have used the data from a 2013 case study provided by Wang et al. [31]. Those case study data have been also used in the later works, so it is very convenient to show the advantages or shortcomings of the proposed model. Because of its completeness and use by other authors the total of eight routes, mostly from the Hong Kong based shipping operator OOCL, has been used and the planning horizon for the fleet deployment problem has been set to six months. Traditionally, the service schedule is based on the weekly calls. The number of voyages required on route $r$ is set to $N_r = 26$.

The data are presented in Table 1:

**Table 1.** Dataset Source [7].

| | **Types of Ships** | | | | |
| --- | --- | --- | --- | --- | --- |
| | t = 1 | t = 2 | t = 3 | t = 4 | t = 5 |
| $V_k$ | 2808 | 3218 | 4500 | 5714 | 8063 |
| $c_{kr}$ | 19.8 | 22.5 | 30.9 | 38.8 | 54.2 |
| $c_k^{out}$ | 1.82 | 2.34 | 3.21 | 4.32 | 5.12 |
| $c_k^{in}$ | 2 | 2.6 | 3.5 | 4.7 | 6 |
| $N_k^{max}$ | 2 | 2 | 9 | 2 | 12 |
| $NCI_k^{max}$ | 10 | 10 | 10 | 6 | 6 |
| $t_{k1}$ | 25.2 | 24.1 | 21.9 | 21.6 | 21.0 |
| $t_{k2}$ | 20.7 | 19.7 | 17.9 | 17.6 | 17.2 |
| $t_{k3}$ | 15.1 | 14.4 | 13.1 | 12.9 | 12.6 |
| $t_{k4}$ | 38.9 | 37.1 | 33.8 | 33.2 | 32.4 |
| $t_{k5}$ | 63.8 | 60.9 | 55.4 | 54.5 | 53.2 |
| $t_{k6}$ | 22.8 | 21.7 | 19.8 | 19.4 | 19.0 |
| $t_{k7}$ | 58.0 | 55.4 | 50.4 | 49.5 | 48.4 |
| $t_{k8}$ | 2.1 | 2.0 | 1.8 | 1.8 | 1.8 |

The demand overflow risk levels on each route $r$ have been set to 1%, 5%, 10% and 15%, respectively.

The mean values of the demand, based on Ng research [7] for given routes $r$ are $\mu_1 = 78,000$, $\mu_2 = 52,000$, $\mu_3 = 52,000$, $\mu_4 = 130,000$, $\mu_5 = 78,000$, $\mu_6 = 52,000$, $\mu_7 = 78,000$ and $\mu_8 = 26,000$ containers. The variance of demands is specified as $\sigma_r = 0.005 \times \mu_r \times \mu_r$, $r \in R$.

For the initial setting purpose, we have assumed that there are no historical data available or that such data are not reliable and hence unusable. Also, another working assumption, generally applicable across the robust optimization is that true distribution is unknown for both MIP and DR ROM. Therefore, each set of cargo demands will have to follow the preset distribution. Means and variances have been specified as stated above.

Based on Monte Carlo sampling method (20 iterations used) a base of fictive historical data has been established to help us obtain empirical mean and covariance (variance included as well). With those two values at hand, it is possible to get optimal values of discrete MIP, as well as DR ROM.

Most importantly, the decision solutions such as $n_{kr}^{own}$, $n_{kr}^{in}$, $x_{kr}$, $z_s^{hod}$ for both discrete and especially for robust models are solvable in acceptable CPU time. This means that using available commercial solvers, such as MATLAB, CPLEX, GUROBI, etc. may provide solutions for robust optimal fleet deployment.

For such purpose the MATLAB integrated CPLEX solver code was created to solve the ROM problem. Figure 2 shows the CPU time for solving the problem and it is visible that the optimal solution for our dataset is found in 12.58 s only 66,850.

```
Branch and bound:

    nodes             total          num int           integer           relative
  explored          time (s)        solutions             fval            gap (%)
       204            00.16                03       4.323460×10⁶          1.533670
       707            00.25                04       4.382740×10⁶          1.293026
       708            00.25                05       4.328360×10⁶          1.284467
      1693            00.44                06       4.382070×10⁶          1.277935
      6745            01.67                07       4.323460×10⁶          1.135574
      9469            01.80                08       4.375760×10⁶          0.940432
      9470            01.81                09       4.323460×10⁶          0.933853
    19,470            03.64                09       4.367140×10⁶          0.867443
    29,470            05.32                09       4.366850×10⁶          0.804469
    37,215            06.61                10       4.366850×10⁶          0.738096
    37,216            06.61                11       4.366850×10⁶          0.731499
    47,216            08.34                11       4.363930×10⁶          0.450333
    57,216            10.01                11       4.363640×10⁶          0.295166
    67,216            11.77                11       4.363640×10⁶          0.094875
    69,330            12.12                11       4.363460×10⁶          0.057979
    71,997            12.58                11       4.363460×10⁶          0.000000

Optimal solution found.
```

**Figure 2.** CPU time for solving the Fleet Deployment problem; CPLEX/MATLAB.

Finally, Tables 2 and 3 show the solutions obtained for optimal fleet deployment, concerning the respective route *r* and ships of type *k*.

**Table 2.** Distribution of owned ship types; Source [CPLEX/MATLAB results].

| Ship Type | Route 1 | Route 2 | Route 3 | Route 4 | Route 5 | Route 6 | Route 7 | Route 8 |
|---|---|---|---|---|---|---|---|---|
| t = 1 (2808 TEU) | 0 | 0 | 1 | 0 | 1 | 0 | 0 | 0 |
| t = 2 (3218 TEU) | 0 | 2 | 0 | 0 | 0 | 0 | 0 | 0 |
| t = 3 (4500 TEU) | 0 | 0 | 1 | 1 | 0 | 0 | 0 | 0 |
| t = 4 (5714 TEU) | 0 | 0 | 0 | 0 | 0 | 0 | 0 | 0 |
| t = 5 (8063 TEU) | 0 | 0 | 0 | 0 | 0 | 0 | 0 | 0 |

**Table 3.** Distribution of chartered ship types; CPLEX/MATLAB results.

| Ship Type | Route 1 | Route 2 | Route 3 | Route 4 | Route 5 | Route 6 | Route 7 | Route 8 |
|---|---|---|---|---|---|---|---|---|
| t = 1 (2808 TEU) | 0 | 2 | 2 | 3 | 0 | 0 | 0 | 0 |
| t = 2 (3218 TEU) | 3 | 2 | 0 | 0 | 5 | 0 | 0 | 0 |
| t = 3 (4500 TEU) | 0 | 0 | 0 | 0 | 0 | 0 | 0 | 0 |
| t = 4 (5714 TEU) | 0 | 0 | 0 | 0 | 0 | 0 | 0 | 0 |
| t = 5 (8063 TEU) | 0 | 0 | 0 | 0 | 0 | 0 | 0 | 0 |

It is, of course, only natural that the distribution of owned and chartered ships differs due to the nature of operational costs involved. Table 4 shows the optimal number of voyages per each ship type.

**Table 4.** Distribution of optimal number of voyages per ship type; Source [CPLEX/MATLAB results].

| Ship Type | Route 1 | Route 2 | Route 3 | Route 4 | Route 5 | Route 6 | Route 7 | Route 8 |
|---|---|---|---|---|---|---|---|---|
| t = 1 (2808 TEU) | 0 | 14 | 24 | 25 | 8 | 26 | 26 | 26 |
| t = 2 (3218 TEU) | 26 | 12 | 0 | 0 | 18 | 0 | 0 | 0 |
| t = 3 (4500 TEU) | 0 | 0 | 2 | 1 | 0 | 0 | 0 | 0 |
| t = 4 (5714 TEU) | 0 | 0 | 0 | 0 | 0 | 0 | 0 | 0 |
| t = 5 (8063 TEU) | 0 | 0 | 0 | 0 | 0 | 0 | 0 | 0 |

To provide useful support to the decision-making process, the implemented algorithms that solve the underlying decision problem must be able to provide solutions to the real-world problem cases at reasonable computational times [34]. Therefore, an effective metaheuristic approach to the size of the ship fleet and the problem of various maintenance operations including fleet deployment is proposed. The metaheuristic approach is a version of the greedy randomized adaptive search procedure—GRASP [22,23].

It consists of the initial feasible solutions to a problem by a randomized algorithm. The solution to the fleet size problem and its combination consists of a viable fleet of maintenance vessels and their respective maintenance bases. After the construction of feasible solutions, the solutions are improved by the local search process [34]. The local version search process performed is a taboo search and explores the neighborhood solutions to the initial solution. The adjacent solutions are defined by several neighborhood operators that define changes in the size of the fleet and a combination of the initial solution. The overall generic pseudo-code for the GRASP procedure is shown in Figure 3.

```
procedure grasp ()
    1       InputInstace ();
    2       for   GRASP stopping criterion not satisfied  ⟶
    3               ConstructGreedyRandomizedSolution (Solution);
    4               LocalSearch (Solution);
    5               UpdateSolution (Solution,BestSoulutionFound);
    6       rof;
    7       return(BestSolutionFound)

end grasp;
```

**Figure 3.** A generic GRASP pseudo-code.

Line 1 represents the problem input and iterations are performed within the code lines from 2 to 6. The criterion for stopping the procedure is set for a certain number of iterations or alternatively, if the solution is found. Line 3 is GRASP construction phase, as shown in Figure 3, while line 4 is the local search phase (see Figure 4). For the case that an improved solution is found, the incumbent is updated in line 5.

Figure 5 represents the search phase of the pseudo code, until the local optimum is found or the preset number of iterations (repetitions) on all available ships is completed [6].

```
procedure ConstructGreedyRandomizedSolution(Solution)
              1 Solution = {};
         2 for Solution construction not done  ———————▶
              3 Make RCL (RCL):
         4 s= SelectElementAtRandom (RCL);
              5 Solution = Solution U {s};
              6 AdaptGreedyFunction (s);
                   7 rof;
```

**Figure 4.** GRASP construction phase of pseudo-code.

```
procedure      local(P,N(P),s)
1              for s not locally optimal  ———————▶
2                       Find a better solution t ∈ N(s);
3                       Let s = t;
4              rof;
5              return(s as local optimal for P)
End local;
```

**Figure 5.** GRASP local search phase of pseudo-code.

## 5. Discussion

While applying the optimization robust model, the costs and fees were calculated. Since there are two types of revenues, the income of chartering out the vessels and the income from the container freight, both are included in the model. A few different types of expenses were taken into account, such as cargo handling costs, vessel operation costs (OPEX), and investments into the chartered vessels. These costs were calculated through the robust optimization model. If we take into consideration the operating costs for ship $k$ on its route $r$, we must mention that $C_{kr}$ has the cost of fuel, cost of daily labor and harbor and light dues. Based on this, the ship OPEX plus charter fees were also calculated in the mode. Based on all the formulations set above, we were able to calculate an optimization model with the objective of maximizing the income, which is shown in Equation (5). This is how an expected value model was obtained. In Equations (6) and (7) it is made sure that the number of ships, both owned and chartered ones, did not exceed the available number of vessels.

Equation (10) represents the total transport capacity of ships deployed on linear routes. It is clear here that the constraint ensured that there is no exceeding of the capacity from the number of containers which are offered for transport.

For the fleet deployment problem (P), the following neighborhood operators (N) are used in the local search process:

- Add a long-term vessel: Add to the solution a single viable vessel for the long-term charter.
- Remove a long-term vessel: Remove from the solution a single long-term chartered vessel.

- Add a vessel for a short time charter: Add to the solution a single viable vessel for the long-term charter, but for a strictly specified charter period.
- Remove vessel short-term: Remove from the solution a single viable vessel for the long-term charter, but for a strictly specified charter period.
- Replace vessels for a short-term 1: Remove one vessel with short-term charter during the charter period and replace it with another type of vessel chartered on a short-term.
- Replace vessels for a short-term 2: Remove one vessel for short-term charter during the charter period and insert it in another charter period.
- Replace vessels: Remove from the solution a vessel with a long-term charter and replace it with another type of vessel on a long-term charter.

The initial feasible solution and its adjacent solutions are called candidate solutions. Each candidate solution is evaluated through a simulation process consisting of a scenario generator that generates a series of time datasets and corrective maintenance task sets (if needed, must be included in the problem formulation) and a calculator that calculates the value of the objective solution function for the candidates for the given time data [24].

The assessment of the candidate's solution is then the average sum of the values of the objective function in all scenarios.

When it comes to the area of robust optimization, the research works in this area [42–47] must be developed to see that there are not too many formal arguments clearly defining the uncertainty set. What had to be done is to use the business intuition combined with the need to adapt the uncertainty set. In this way it would be possible to solve the problem in a reasonable time. It was also necessary to establish what kind of robustness and uncertainty sets will be used to solve the problem. What was crucial for this problem is that different uncertainty sets were defined.

Speaking of the integrated machine learning and metaheuristics, there is a line that must be explored to get the best results. This line refers to the use of general scheme of integration by using meta-learning techniques [48]. Since the algorithm already exists, a mechanism that chooses the best algorithm is used to obtain the results and achieve convergence.

## 6. Conclusions

The paper deals with the improved approach of approximations related to the problem of fleet deployment under a set of uncertainties. The target was to research the possibilities of application of robust optimization model in case of container fleet management to optimize the fleet capacities and expenses. It was therefore, necessary to formulate a model of robust optimization with the corresponding sets of uncertainties, and to make a metaheuristic analysis for such a model. In modelling process transformation of the mixed integer according to the distribution robust chance constraints was performed.

The key output element is to guarantee a certain level of shipping service that is measured by the risk of excessive demand. Theoretically, it has been shown that the new formulation has better performance than the most modern models known to the authors, which has been proven by numerical experiments. By using such a model, it is possible to achieve better fleet use with lower costs and optimization of fleet deployment and capacity use.

Future directions of research may include the improvement of approximation methods, because there are still certain gaps between reliability outside the sample and the given level of risk. Therefore, more advanced approximation approaches should be adopted. In addition, randomly distributed, distribution robust programming can be further applied to other areas, such as the problem of parcel distribution, or the problem of real-time vehicle movement.

**Author Contributions:** M.B.: Conceptualization, Methodology, Validation, Supervision. K.R.: Conceptualization, Methodology, Validation, Supervision. V.J.: Conceptualization, Methodology, Software. All authors have read and agreed to the published version of the manuscript.

**Funding:** This research was funded by the Faculty of Transportation and Traffic Sciences, University of Zagreb.

**Institutional Review Board Statement:** Not applicable.

**Informed Consent Statement:** Not applicable.

**Data Availability Statement:** CCX: Ningbo–Shanghai–Pushan–Los Angeles–Oakland–Pushan–Dalian-Xingang–Qingdao, https://www.oocl.com/SiteCollectionDocuments/OOCL/Our%20Services/Service%20Routes/TPT/CCX_Mar10.pdf (accessed on 10 October 2021). CPX: Shanghai–Ningbo–Shekou–Singapore–Karachi–Mundra–Port Klang–Singapore–Hong Kong, https://www.ocean-insights.com/liner-news/enhancements-cix1-pmx-products (accessed on 10 October 2021). GIS: Port Klang–Nhava Sheva–Karachi–Bandar Abbas–Jebel–Mundra–Cochin–Signapore. IAX: Port Qasim–Nhava Sheva–Mundra–Damietta–New York–Norfolk–Savannah–Charleston-Port Said–Jeddah, https://www.apl.com/products-services/line-services/flyer/ (accessed on 10 October 2021). INDAMEX NCE: Shanghai–New York–Norfolk–Savannah-Pushan-Qingdao-Ningbo-Shanghai, https://www.oocl.com/SiteCollectionDocuments/OOCL/Our%20Services/Service%20Routes/TPT/NCE_Apr10.pdf, (accessed on 10 October 2021). UKX: Southapton–Hull–Grangemouth NZX: Port Klang–Singapore–Brisbane–Auckland–Lyttelton–Wellingtono–Napier–Tauranga–Brisbane–Port Klang, http://www.apl.com/products-services/line-services/flyer/KIXANL (accessed on 10 October 2021). SCE: Shekou-Hong Kong-New York-Norfolk-Savannah–Kaohsiung, https://www.oocl.com/china/schi/localinformation/localnews/2007/Pages/29May0701a.aspx?site=china&lang=schi. (accessed on 10 October 2021).

**Conflicts of Interest:** The authors declare that they have no known competing financial interest or personal relationships that could have appeared to influence the work reported in this paper.

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
