# Peer review of "Distributionally Robust Model and Metaheuristic Frame for Liner Ships Fleet Deployment"

_sustainability, doi:10.3390/su14095551_

Round 1

Reviewer 1 Report

This paper presented the “Distributionally robust model and metaheuristic frame for liner ships fleet deployment”. The shipping industry is very important for the global trade in the world. However, it is still under the heavy enormous pressure and environmental changes. The aim of this study is to provide a model for minimizing costs while managing cargo demand risk. The contribution is offer a model for deploying a fleet under certain demand in shipping closer to real living conditions.  This paper is relevant and valuable to the readers of the Sustainability. To guide you, I have several suggestions, which I believe would improve your manuscript.

  1. Introduction, a significant weakness of this paper is the insufficient explanation for why the research interest is specially placed in the container shipping industry.  In this paper, the authors described the difficulties and challenges the container shipping companies have encountered when trying to enforce uncertain environmental environment and cost. However, it seems to me that those challenges are universal across many industries, not just the shipping businesses. The authors need to improve the motivation of their study. I believe that several parts of the paper including in the introduction need to be rewritten and improved. 
  2. The value of this paper is not significant. This is mean that the authors shall demonstrate what is difference between this study and the others. The authors may read more papers in this field and do dome example to explain the differ from the other researches to increase research values.
  3. In page 1, line 36 to 40, “Global world trade has reached in 2019……global maritime transportation volume”, it looks like research background. The authors might consider shift them to the “Introduction” for enhancing the background.
  4. The introduction needs to motivate the research with appropriate research questions, examples and literature. The literature review section needs to thoroughly discuss theory and constructs that build the theoretical models. Precision in the application of theory to create hypotheses needs improvement.
  5. Figure 1 is not clear. Please provide a better one in the revising version.
  6. The conclusions drawn in this paper are too simple. They are just some results presentation, but they have not been excavated more deeply. Relevant conclusions are difficult to apply to the issue.

Author Response

Dear reviewer, there is a revised version of the paper attached.

Reviewer 2 Report

-----------------------

Abstract
It would be helpful to rewrite the Abstract to answer three key questions:
1. Which issue/problem is addressed in this article?
2. What solution do we offer (describe our contribution)?
3. What are the properties of our solution.?

Answers should contain two or three sentences and should be balanced in content.  

-----------------------

Check your grammar. The very first sentence says ". . . and infrastructure . . . "

-----------------------

You write below ". . . optimal use of shipping . . . ". All real problems are NP-complete and never have to do with "optimal use." They are only approximations of the optimum.

-----------------------

In general, such problems can be solved in the following ways:
1. Find the optimal solution (which is an NP-hard problem and it is not for debate)
2. Use of heuristics
3. Use of simulations (which are increasingly used nowadays of increased computational power - this approach includes the Monte Carlo method)
4. Mixture of points 2 and 3

Before the "Model construction" section, let us know what approach you use and how - just a sentence or two. 

-----------------------

In the "Model construction" section, write precisely and explicitly wherein the model your contribution is (see also the debate of Abstract). What you have done in the model is different from what is already published (what already exists).

-----------------------

In line 427, you wrote, "The diagram ... Figure 5". But Figure 5 does not represent any diagram.

-----------------------

Author Response

(The authors gave the same response as above.)

Round 2

Reviewer 1 Report

This paper presented the “Distributionally robust model and metaheuristic frame for liner ships fleet deployment”. The shipping industry is very important for the global trade in the world. However, it is still under the heavy enormous pressure and environmental changes. The aim of this study is to provide a model for minimizing costs while managing cargo demand risk. The contribution is offer a model for deploying a fleet under certain demand in shipping closer to real living conditions.  This paper is relevant and valuable to the readers of the Sustainability.

I appreciate authors' efforts on revising the manuscript. There is a small concern that the Figure 2 is not clear. Please try to keep a new one clear for reading friendly. Thanks.

Author Response

Dear reviewer, thanks for your comments and suggestions. Well noted.

The figure 2 has been improved to show better clarity. Once again, paper has been checked for grammar, style and syntax and minor improvemets have been done to the text.

Hope will sufice.

Best regards, authors.